# An Entropy-Based Approach to Model Selection with Application to Single-Cell Time-Stamped Snapshot Data

**DOI:** 10.3390/e27030274

**Published:** 2025-03-06

**Authors:** William C. L. Stewart, Ciriyam Jayaprakash, Jayajit Das

**Affiliations:** 1GIG Statistical Consulting LLC., 391 E. Livingston Avenue, Columbus, OH 43215, USA; 2Department of Physics, The Ohio State University, Columbus, OH 43210, USA; 3Steve and Cindy Rasmussen Institute for Genomics, Abigail Wexner Research Institute at Nationwide Children’s Hospital, Columbus, OH 43205, USA; 4Department of Pediatrics, College of Medicine, The Ohio State University, Columbus, OH 43210, USA

**Keywords:** single-cell single-cell protein expression, cross-entropy, cytometry by time of flight (CyTOF), time stamped snapshot data, generalized method of moments, model selection, natural killer cell, Akaike information criterion, Kullback–Leibler divergence

## Abstract

Recent single-cell experiments that measure copy numbers of over 40 proteins in thousands of individual cells at different time points [time-stamped snapshot (TSS) data] exhibit cell-to-cell variability. Because the same cells cannot be tracked over time, TSS data provide key information about the statistical time-evolution of protein abundances in single cells, information that could yield insights into the mechanisms influencing the biochemical signaling kinetics of a cell. However, when multiple candidate models (i.e., mechanistic models applied to initial protein abundances) can potentially explain the same TSS data, selecting the best model (i.e., model selection) is often challenging. For example, popular approaches like Kullback–Leibler divergence and Akaike’s Information Criterion are often difficult to implement largely because mathematical expressions for the likelihoods of candidate models are typically not available. To perform model selection, we introduce an entropy-based approach that uses split-sample techniques to exploit the availability of large data sets and uses (1) existing generalized method of moments (GMM) software to estimate model parameters, and (2) standard kernel density estimators and a Gaussian copula to estimate candidate models. Using simulated data, we show that our approach can select the ”ground truth” from a set of competing mechanistic models. Then, to assess the relative support for a candidate model, we compute model selection probabilities using a bootstrap procedure.

## 1. Introduction

Ordinary differential equations (ODEs) are commonly used to model the sub-cellular dynamics of proteins and mRNA [1,2,3]. Usually, ODEs describe the deterministic dynamics of average concentrations, which facilitates the estimation of reaction rates (*also known as* model parameters) from experimental data. This is often an important step towards building mechanistic biological models, but the task of estimating model parameters from experimental data is challenging for a variety of reasons [4,5,6], especially when the number of distinct proteins measured in experiments is smaller than the number of model parameters. Recent developments in single-cell experimental techniques for the longitudinal measurements of transcripts and proteins in an individual cell, such as single-cell RNA-seq [7] or single-cell mass cytometry by time-of-flight (CyTOF) [8,9], which can simultaneously measure over a thousand different RNA sequences or more than thirty different protein species in a single cell, appear to alleviate this problem. Since individual cells are not tracked across time in these experiments, the measurements generate a large collection of time-stamped snapshot (TSS) data. Another challenge stems from the cell-to-cell differences in the copy number of a protein (or abundance), which contains variation present at the pre-stimulus state (*also known as* extrinsic noise), as well as variation arising from the inherent stochasticity of biochemical reactions over time (*also known as* intrinsic noise) [10,11]. When the observed protein abundances are large, intrinsic noise can be ignored, extrinsic noise is known to play a significant role [12], and single-cell protein signaling kinetics can be well approximated by ODEs. By comparing the distribution of protein abundances in single cells observed at time *t* to the predictions obtained from an ODE or a stochastic model that evolves single-cell protein abundances seen at an earlier time (e.g., t=0, *also known as* initial conditions), we can estimate the parameters of the candidate model using a generalized method of moments (GMM) approach [13,14]. GMM, which contrasts sample moments and their corresponding expectations, is widely used in econometrics [15]. In this paper, we propose an entropy-based approach to address the larger question of model selection for systems that can be described by deterministic dynamical models (e.g., sets of ODEs) with randomness arising from initial conditions. Specifically, for the models considered here, we show that our cross-entropy [16] approach can find the *best* ODE model from a set of competing candidate models, where the best model neither over-fits nor under-fits the available data.

The primary goal of model selection is to find the best model relative to some defensible criterion, and two attractive criteria are cross-entropy and Kullback–Leibler (KL) divergence [17]. The latter is a non-negative number that, for a pair of random variables can provide a useful measure of dependence known as mutual information [18]. But more generally, KL divergence measures the “distance” between two probability distributions [13] where the KL divergence vanishes for a pair of identical distributions. Usually, the distribution that gave rise to the protein abundances observed at time *t* (denoted *f*) is considered to be the “ground truth”, while the other distribution is most often a candidate distribution (or model) that is presumed to be “close” to *f*. A defining feature of KL divergence is that it is zero if and only if the candidate model and *f* are the same. Typically, the best candidate model will strike a balance between under-fitting (i.e., over-estimating the random error) and over-fitting (i.e., under-estimating the random error). Since KL divergence is a difference in expectations taken with respect to *f* (i.e., cross-entropy minus entropy), and since entropy depends only on *f*, it suffices (for the purpose of model selection) to find the model with the smallest cross-entropy. Given a finite set of candidate models, the model with the smallest cross-entropy is also the model with the smallest KL divergence, which makes it the best approximating model to *f*.

There are however two very important challenges when performing model selection from TSS data. First, the multivariate probability density of the observed protein abundances at time *t* (denoted symbolically as *f*, and in words as the “ground truth”) is rarely known. Second, while we can estimate the parameters of each candidate model, we cannot evaluate the likelihood of any model. Fortunately, we can deal with the first challenge by minimizing cross-entropy (instead of KL divergence), as cross-entropy only requires *realizations* from *f*, not complete knowledge of *f*. As for the second challenge, which cannot be avoided, we tackle it head-on by estimating the likelihood for each candidate model (see Section 2 and Appendix B for more details) from the time evolution of initial conditions (see Section 3 and Appendix A for more details).

The remaining sections of this paper are organized as follows. In the first subsection of Methods, we give mathematical descriptions of KL divergence and cross-entropy, and we briefly explain how parameters of a candidate model are estimated. Then, in the second subsection of Methods, we outline our model selection approach, leaving technical details (such as the estimation of the Gaussian copula, marginal densities, and marginal cumulative distribution functions) to Appendix A and Appendix B. Furthermore, in the second subsection of Methods, we briefly describe a complementary model selection approach that applies Akaike Information Criterion [19] corrected for small samples (AICc) [20] to differences between the mean protein abundances observed at time *t* and the mean protein abundances predicted at time *t*. In Section 3, we describe how synthetic TSS data are generated, and we describe the time-evolution of initial conditions using different candidate models. Finally, we demonstrate the utility of our model selection approach in Section 4, and we give some interesting insights and suggestions for further improvements in Section 5.

## 2. Methods

### 2.1. Kullback–Leibler and Cross-Entropy

Consider the following: from a large collection of genetically identical cells [e.g., a clonal population of Natural Killer (NK) cells], a subpopulation is extracted, and the abundances of *n* distinct proteins are observed in each cell. We denote a random draw from this subpopulation by x[0]. Now, cells obtained from a different subset of the original collection are allowed to evolve over time. For these cells, which are observed for the same set of distinct proteins at time *t*, let y[t] denote a random draw from this subpopulation. Collectively, the protein abundances in cells observed at time 0 and in cells observed at time *t* are a simple example of TSS data.

We assume that x[0]∼G and that y[t]∼F with density f(y[t]) and that y[0] (which is not observed) has the same distribution as x[0], since they both represent random draws from the same initial collection of similar cells. Furthermore, in accordance with most real-world applications, we assume that *G*, *F*, and *f* are almost never known exactly, (i.e., mathematical expressions are typically not known, but empirical estimates from observed data may be possible). Now, consider modeling the time-evolution of TSS data with different deterministic models $d_{i}$ for *i* = 1, 2, and 3, where each $d_{i}$ is a set of coupled nonlinear ODEs descibing the dynamics of single-cell protein abundances over time. Specifically, the *i*th model $d_{i}$ has a set of reaction rates (see Figure 1) that depend on freely varying parameters $\theta_{i}$ (see Appendix A for more details). We evolve initial conditions x[0] to time *t* using ODE model $d_{i}$ to arrive at predicted abundances $x_{i}\left[t\right]$ which have distribution $H_{i}$ and density $h_{i}$. We refer to $h_{i}$, interchangeably, as the ith candidate model. For each candidate model, we assume there exists a unique parameter vector $\theta_{i}^{*}$ that minimizes(1)$$\begin{array}{c} E_{f}\log\frac{f\left(y[t]\right)}{h_{i}\left(y[t]|\theta \right)}\;, \end{array}$$
where the expectation in expression (Equation 1) is the Kullback–Leibler (KL) divergence between f(y[t]) and candidate model $h_{i}\left(y\left[t\right]|\theta \right)$. Note that, when the ith candidate model and the “ground truth” are the same, $h_{i}=f$ and the KL divergence is zero. Since we have restricted attention to a single point in time, we suppress the dependence of $x_{i}\left[t\right]$ and y[t] on *t* hereafter.

KL divergence can also be expressed as the difference between cross-entropy and entropy(2)$$\begin{array}{c} -E_{f}\log h_{i}\left(y|\theta \right)-\left[-E_{f}\log f\left(y\right)\;\right]. \end{array}$$So, when $\theta =\theta_{i}^{*}$, Expression (Equation 2) is the KL divergence between *f* and $h_{i}$. Since entropy does not depend on $h_{i}$, it suffices (for the purpose of model selection) to find the candidate model that minimizes cross-entropy:(3)$$\begin{array}{c} CE\left(f\left(y\right)\|h_{i}\left(y|\theta_{i}^{*}\right)\right)\equiv -E_{f}\log h_{i}\left(y|\theta_{i}^{*}\right)\;. \end{array}$$However, because *f* is rarely known, the cross-entropy in Equation (Equation 3) cannot be computed. Instead, we must approximate the cross-entropy by averaging over N independent realizations of *f*. To compute the approximation, we also need $\theta_{i}^{*}$ and $h_{i}$, but unfortunately, both are rarely known as well. In the case of $\theta_{i}^{*}$, we can replace it with the generalized method of moments (GMM) estimate of θ (denoted ${\tilde{\theta }}_{i}$). Briefly as in [13], we define ${\tilde{\theta }}_{i}$ to be the value of θ that minimizes the weighted sum of squared differences between the first and second moments of the observed abundances *y*, and the first and second moments of the predicted abundances $x_{i}$ [15]. More generally, if one thinks of the differences between corresponding moments as a difference vector, then the quadratic form of this vector and a symmetric, positive-definite weight matrix (denoted *W*) defines a norm, and this norm is the cost that ${\tilde{\theta }}_{i}$ minimizes. There are many possible choices for *W* (see [14,15,21] for more details). Note that for most over-determined systems, ${\tilde{\theta }}_{i}$ is consistent for $\theta_{i}^{*}$ [22]. In the next subsection, we will describe how $h_{i}$ can be estimated from initial conditions x[0], coupled ODEs $d_{i}$, and its corresponding estimated reaction rates ${\tilde{\theta }}_{i}$. Our estimate of $h_{i}$ is denoted ${\hat{h}}_{i}\left(y|{\tilde{\theta }}_{i}\right)$.

### 2.2. Computing Approximate Cross-Entropy

Single-cell experiments such as CyTOF typically measure protein abundances across thousands of cells, and as such, TSS data often contain a wealth of information for estimating marginal densities and marginal distribution functions. Consequently, we decided to leverage Sklar’s theorem [23] to estimate the multivariate density $h_{i}\left(y|{\tilde{\theta }}_{i}\right)$ from its marginal densities, marginal distribution functions, and a copula describing the dependence between the abundances of *n* distinct proteins in a single cell (see Appendix B for more details). Here, we implement a Gaussian copula, which is computationally fast, mathematically convenient, and (at least in the case of TSS data) quite accurate (see Appendix C).

Another benefit of working with large samples is that split-sample techniques allow us to avoid the bias that typically arises when parameter estimation and model selection are performed on the same dataset [19]. Specifically, we use 20% of the available TSS data for parameter estimation (i.e., to compute ${\tilde{\theta }}_{i}$), and 80% for model selection (i.e., to compute ${\hat{h}}_{i}\left(y|{\tilde{\theta }}_{i}\right)$. Now, the cross-entropy that we want (see Equation (Equation 3)), can be approximated by(4)$$\begin{array}{c} -E_{f}\log{\hat{h}}_{i}\left(y|{\tilde{\theta }}_{i}\right) \end{array}$$
and the expression in (Equation 4) can be estimated from the corresponding sample average (denoted ACE):(5)$$\begin{array}{c} ACE\left(f\left(y\right)\|{\hat{h}}_{i}\left(y|{\tilde{\theta }}_{i}\right)\right)\equiv \frac{1}{N}\sum_{k}-\log{\hat{h}}_{i}\left(y_{k}|{\tilde{\theta }}_{i}\right), \end{array}$$
where N independent cells at times 0 and *t* are used to compute ${\hat{h}}_{i}\left(y|{\tilde{\theta }}_{i}\right)$ and ACE, respectively.

Because our proposed model selection approach is based on an approximation, it is useful to have (1) independent confirmation that the selected model *actually* minimizes KL divergence, and (2) some measure of relative support for the model that ACE selects compared to the other models that could have been chosen. To address the first point, we compute AICc for each candidate model by working with the likelihood for the mean difference. Specifically, we assume that the mean of *y* minus the mean of $x_{i}$ is multivariate normal with expectation zero, and that its variance–covariance matrix has off-diagonal elements equal to zero [20]. Because (1) the initial protein abundances are independent across cells, (2) we assume initial abundances are also independent within cells for simplicity, and because (3) the sample size is large, departures from normality and departures from independence should be small. Note that, when model selection is based on a transformation of the original TSS data (e.g., the mean difference), it is usually possible to construct several different AICc-like statistics. The one we chose to compute here is easy to implement and natural, but more elaborate constructions are also possible. When the candidate model that minimizes ACE also minimizes approximate AICc, one has (to some degree) additional assurance that the selected model minimizes cross-entropy [see Equation (Equation 3)], and therefore minimizes KL divergence [see Equation (Equation 2)]. Furthermore, by bootstrapping the observed TSS data, we can estimate the *model selection* probabilities for each candidate model (i.e., the probability that model $h_{i}$ is selected). Typically, model selection probabilities will help users quantify and interpret the level of relative support for the selected model in ways that are comparable to, or better than, differences in AIC or AICc [24].

## 3. Data Description

In most real-world applications that model the time-dependent kinetics of protein abundances, the “ground truth” (denoted *f*) is rarely known. However, because we are primarily concerned with improving model selection (i.e., increasing accuracy, increasing usability, and developing additional measures of relative support), we chose to mimic observed TSS data *y* by simulating (without intrinsic noise) protein abundances at time *t* from three scenarios of interest: SMALL, MEDIUM, and LARGE (see Table A1 for more details). For the SMALL, MEDIUM, and LARGE scenarios, we create ground truth models where we vary one, two, and three parameters of θ. For instance, in the ground truth MEDIUM scenario, candidate model $h_{2}$ is expected to explain the TSS data better than $h_{1}$, which is too simplistic, and better than $h_{3}$, which over-fits to noise in the data (see Table 1).

For each “ground truth” scenario, we always know which candidate model should yield the best fit. For example, consider candidate model $h_{2}$ which has two freely varying parameters, $\theta_{1}$ and $\theta_{3}$, because we set $\theta_{2}=9\times \theta_{1}$. Relative to $h_{1}$ and $h_{3}$, candidate model $h_{2}$ should be the *closest* to the “ground truth” MEDIUM scenario, provided that under-fitting and over-fitting are accounted for appropriately (see Table 2). Similarly, candidate models $h_{1}$ and $h_{3}$ are expected to to be *closest* to “ground truth” scenarios SMALL and LARGE, respectively, since in $h_{1}$ we set $\theta_{2}=9\times \theta_{1}$ and $\theta_{3}=2\times \theta_{1}$ with only $\theta_{1}$ varying freely; and since $h_{3}$ has all three parameters varying freely. Of course, when implementing ACE (and AICc), we pretend that the “ground truth” is unknown.

**Table 1 entropy-27-00274-t001:** **Under-fitting and over-fitting.** The cost (shown in brackets) decreases as the complexity of the candidate model increases, so $h_{3}$ has the lowest cost. Yet ACE selects $h_{2}$, the correct candidate model, 95% of the time (see Table 3), which implies that ACE appropriately balances the under-fitting of $h_{1}$ and the over-fitting of $h_{3}$. The parameter estimates $\theta \equiv (\theta_{1},\;\theta_{2},\;\theta_{3})$ are shown for each candidate model; and as a point of reference, the “ground truth” MEDIUM scenario is shown in bold.

Model:	$\theta_{1}$	$\theta_{2}$	$\theta_{3}$	Cost
MEDIUM:	**0.10**	**0.90**	**0.18**	[–NA–]
$h_{1}$:	0.093	(9 × 0.093)	(2 × 0.093)	[0.0370]
$h_{2}$:	0.100	(9 × 0.100)	0.182	[0.0074]
$h_{3}$:	0.097	0.896	0.182	[0.0071]

To generate TSS data, we begin by simulating uncorrelated initial conditions from a multivariate log-normal distribution with parameters μ = (5.25, 7.60, 5.25, 7.60, 5.25, 5.25), and $\sigma^{2}$ = (0.15, 0.06, 0.15, 0.06, 0.15, 0.15). Now, let us consider simulating data for the “ground truth” LARGE scenario. To accomplish this, we take half of the initial conditions (discussed immediately above) and we evolve them to time *t* using coupled ODEs $d_{3}$ and parameters (0.10, 0.95, and 0.18). Computing ACE and model selection probabilities for all three candidate models takes about 2 h on a 2.5 GHz computer, but apart from computational time, there is no limit on the number of candidate models one can consider.

## 4. Results

We show that our model selection approach works (Table 2) and that approximate AICc often selects the same candidate model as ACE (Table 3). Next, we repeated our model selection approach for each of 1000 bootstrap resamples of the simulated TSS data, and we computed model selection probabilities for each candidate model.

For the “ground truth” LARGE scenario, ACE selected the correct candidate model $h_{3}$ (with 3 freely varying parameters) 100% of the time. Similarly, for the “ground truth” MEDIUM and SMALL scenarios, ACE selected the correct candidate models $h_{2}$ (with 2 freely varying parameters) and $h_{1}$ (with 1 freely varying parameter) 95% and 76% of the time, respectively (see Table 3).

## 5. Discussion

### 5.1. New Tools for Model Selection with Single-Cell Data

Mechanistic models based on ODEs describing subcellular kinetics of proteins are widely used in computational biology for gleaning mechanisms and generating predictions. On the other hand, to model random gene expression data, stochastic mechanistic models such as the telegraph model have been used [25,26]. It is common to have multiple candidate mechanistic models that can be set up to probe different hypotheses describing the same biological phenomena, and an important task in model development is to rank order the candidate models according to their ability to describe the measured data. The availability of large, high-dimensional, single-cell datasets allows for estimation of model parameters using mean values and higher-order moments of the measured data; however, rank ordering candidate models from such data may not be straightforward when using standard approaches in model selection (e.g., AIC, AICc, and Kullback–Leibler). Here we propose a model selection approach based on cross-entropy for ODE-based models that are calibrated against means and higher-order moments of the measured data. We show as “proof-of-concept” that our proposed approach successfully rank orders a set of ODE models against synthetic single-cell datasets.

### 5.2. ACE Complements Approximate AICc

As shown in Figure 2 (Panel C), approximate AICc is virtually independent of ACE, and concordance between ACE and approximate AICc appears to provide additional support for the candidate model selected by ACE. So, any improvement in approximate AICc would likely only benefit ACE. One potential area for improvement might be a recalibration of the penalties used in approximate AICc. In particular, it’s not immediately clear what the penalty should be for approximate AICc, as the parameter estimation is based on thousands of cells, whereas the model selection is based on differences in only six means. Moreover, when one combines a split-sample approach with model selection (as we have done here), the penalty terms in AIC and AICc may no longer be needed [27]. Indeed, the so-called *penalties* are actually *corrections* for the bias that arises when parameter estimation and model selection are performed on the same dataset.

By design, the additional penalty term in AICc depends on the corresponding sample size (here, AICc uses only six means). As such, the number of distinct proteins must be larger than the number of freely varying parameters; otherwise, the denominator of the additional penalty term will be zero or negative. Also, because the AICc likelihood is base on the differences in means, AICc may have more difficulty than ACE accurately rank ordering two or more candidate models with similar means. Note that ACE (as implemented here, with a split-sample approach) does not have either design limitation: (1) ACE penalizes indirectly for complexity and does not require a bias correction, and (2) ACE makes use of the entire multivariate density, not just the means.

For the MEDIUM and LARGE “ground truth” scenarios, ACE outperforms approximate AICc, but for the SMALL “ground truth” scenario, approximate AICc does better than ACE. This suggests that the means tend to carry the bulk of the information in the “ground truth” SMALL scenario, so there’s very little benefit to estimating the other candidate models (which contain information about higher-order moments and cross-moments). However, when the higher-order moments and cross-moments begin to matter, as is likely the case with the more complex “ground truth” scenarios, the benefit of estimating the candidate models with 2 and 3 freely varying parameters is likely greater.

### 5.3. Limitations and Future Directions

Presently, our ACE approach to model selection based on TSS data has three main limitations: (1) it is not designed to handle intrinsic noise, (2) there is considerable latitude in terms of multivariate density estimation that we have only scratched the surface of here, and (3) there may be more efficient ways to incorporate split-sample techniques. Extending our ACE approach to include *both* extrinsic and intrinsic noise may be possible for relatively short evolution times and/or for networks with a relatively small number of interacting proteins. Further, for other applications, users may want to include higher-order moments and/or different copulas or kernel density estimators [28,29].

When researchers are unable to specify the full candidate model, the likelihood is not known and model selection is often challenging. However, when the sample size is large and consistent estimators of the model parameters and candidate models exist, we propose a split-sample entropy-based approach that allows users to find the best approximating model to the “ground truth”. Furthermore, our approach is quite flexible with respect to (1) parameter estimation (e.g., choosing which moments to use—first moments only, first and second moments, etc.), and (2) multivariate density estimation (e.g., choosing a “good” kernel density estimator and/or copula for each candidate model).

## 6. A Selected Sensitivity Analysis

To examine what could happen with the analysis of TSS data observed at a different time point (e.g., *t* = 4.5) and at a smaller sample size (e.g., 4000), we show the results below in Table 4 and Table 5, respectively. As expected, with longer evolution times, each of the candidate models are more differentiated, and so selecting the best model is (in some ways) easier. However, relative to *t* = 1.5, the dependence between variables at *t* = 4.5 is increased. As such, the number of *effectively* independent variables [30,31] is reduced at *t* = 4.5, and this makes parameter estimation more challenging, especially when 3 reaction rates are allowed to vary freely. As expected, the performance of ARE drops due to the reduction in sample size.

## Figures and Tables

**Figure 1 entropy-27-00274-f001:**
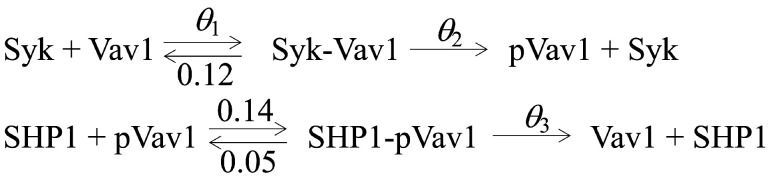
**Minimal NK Cell Signaling Model** shows the biochemical reactions for the six protein species interaction model: Syk, Vav1, Syk-Vav1, SHP1, and SHP1-pVav1. The corresponding set of coupled ODEs uses the reactions shown (above) to explain the deterministic mass action kinetics of this signaling model. To ensure that $\theta =(\theta_{1}$, $\theta_{2}$, $\theta_{3})$ is identifiable, the other three reaction rates were fixed at $0.12s^{-1}$, $0.14s^{-1}$, and $0.05s^{-1}$, respectively.

**Figure 2 entropy-27-00274-f002:**
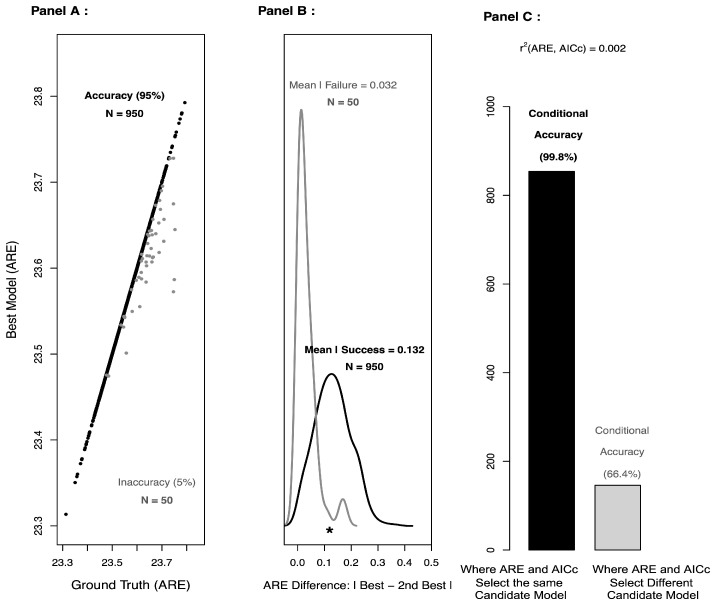
**A detailed examination of the “ground truth” MEDIUM scenario**. Based on 1000 bootstrap resamples, Panel (**A**) shows the overall accuracy (black dots) and inaccuracy (grey dots) of ACE for the “ground truth” MEDIUM scenario. Panel (**B**) shows the distribution of absolute differences between the smallest and second smallest ACE scores when ACE selects the correct model (black) or an incorrect model (grey). The asterisk is the absolute difference between the ACE of $h_{2}$ and $h_{3}$ for data simulated with the “ground truth” MEDIUM scenario. Panel (**C**) shows how concordance with AICc provides additional evidence that the model selected by ACE is correct.

**Table 2 entropy-27-00274-t002:** **Model selection based on a single TSS Dataset.** For each “Ground Truth” scenario (LARGE, MEDIUM, and SMALL), we generated synthetic time-stamped snapshot (TSS) data for 6 proteins across 8000 cells at time t=1.5 s. Columns 2 and 3 show the minimum ACE and the second smallest ACE, respectively. Columns 4 and 5 show the minimum AICc and the second smallest AICc, respectively. The minimization is taken over all three candidate models. The selected candidate model, $h_{1}$, $h_{2}$, or $h_{3}$, is shown in brackets. For both ACE and AICc, the correct candidate model (bold) was selected.

Ground Truth	Minimum ACE	2nd Smallest ACE	Minimum AICc	2nd Smallest AICc
LARGE	23.8 [$h_{3}$]	24.2 [$h_{2}$]	33 [$h_{3}$]	54 [$h_{2}$]
MEDIUM	22.5 [$h_{2}$]	23.6 [$h_{3}$]	20 [$h_{2}$]	32 [$h_{3}$]
SMALL	23.5 [$h_{1}$]	23.6 [$h_{2}$]	10 [$h_{1}$]	21 [$h_{2}$]

**Table 3 entropy-27-00274-t003:** **Accuracy of ACE model selection.** For each “Ground Truth” scenario, 1000 bootstrap resamples were generated. Here, we show the probability of selecting candidate models $h_{3}$, $h_{2}$, and $h_{1}$ using ACE when the “Ground Truth” scenario is LARGE, MEDIUM, and SMALL, respectively. Corresponding probabilities for AICc are given in parentheses.

Ground Truth	$P_{\mathrm{select}}\left(h_{3}\right)$	$P_{\mathrm{select}}\left(h_{2}\right)$	$P_{\mathrm{select}}\left(h_{1}\right)$
LARGE	**1.0 ** (0.68)	0 (0.32)	0 (0)
MEDIUM	0.05 (0.11)	**0.95** (0.89)	0 (0)
SMALL	0 (0.01)	0.24 (0.13)	**0.76** (0.86)

**Table 4 entropy-27-00274-t004:** **Accuracy of ACE model selection (t = 4.5 s).** For each “ground truth” scenario with protein abundances observed across 8000 cells, 100 bootstrap resamples were generated, and the probability of selecting candidate models $h_{3}$, $h_{2}$, and $h_{1}$ is shown. Model selection probabilities for ACE are bold, and AICc is shown in parentheses.

Ground Truth	$P_{\mathrm{select}}\left(h_{3}\right)$	$P_{\mathrm{select}}\left(h_{2}\right)$	$P_{\mathrm{select}}\left(h_{1}\right)$
LARGE	**1.0** (1.0)	0 (0)	0 (0)
MEDIUM	0.05 (0.02)	**0.95** (0.98)	0 (0)
SMALL	0 (0)	0 (0)	**1.0** (1.86)

**Table 5 entropy-27-00274-t005:** **Accuracy of ACE model selection with 4000 cells.** For each “ground truth” scenario observed at *t* = 1.5, 1000 bootstrap resamples were generated, and the probability of selecting candidate models $h_{3}$, $h_{2}$, and $h_{1}$ is shown. Model selection probabilities for ACE are bold, and AICc is shown (without a penalty) in parentheses.

Ground Truth	$P_{\mathrm{select}}\left(h_{3}\right)$	$P_{\mathrm{select}}\left(h_{2}\right)$	$P_{\mathrm{select}}\left(h_{1}\right)$
LARGE	**1.0** (0.65)	0 (0.35)	0 (0)
MEDIUM	0.25 (0.35)	**0.75** (0.65)	0 (0)
SMALL	0 (0)	0.42 (0.33)	**0.58** (0.67)

## Data Availability

The software will be made available to interested researchers upon request (email: minitether@gmail.com).

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
