# Peer review of "An Entropy-Based Approach to Model Selection with Application to Single-Cell Time-Stamped Snapshot Data"

_entropy, 2025, doi:10.3390/e27030274_

Round 1

Reviewer 1 Report

Comments and Suggestions for Authors

In the paper by William CL Stewart et al., authors introduced an entropy-based approach to select models by fitting single-cell protein abundances data. They showed that the approach can outperform the classical AIC and AICc in most situations, and also the approach avoids some limitations of AICc such as the relatively large number of protein copy numbers. Overall, the manuscript is well written and the topic is very important in the field of system biology. 

I have some suggestions. The manuscript tested the approach with only 1000 sample size and a single time point 1.5. It should test how the approach performs when data are measured under different sample sizes and multiple time points. Moreover, the manuscript should add a figure that shows how the data are fitted by different models. Also, there should be some brief discussions on model selection based on the other single-cell expression data (such as distribution data of gene products copy numbers: Mathematical Biosciences 345 (2022) 108780, PLoS Comput Biol 20(5): e1012118)

A typo: Line 66 “minus entropy). and since entropy depends only on f, it suffices (for the purpose of model”, period should be deleted. 

Author Response

Comment 1 - The manuscript tested the approach with only 1000 sample size and a single time point 1.5. It should test how the approach performs when data are measured under different sample sizes and multiple time points.

Response to Comment 1 - Reviewer 1 makes a good point; nevertheless, a touch of clar- ification is probably needed before giving our full response. First, 1000 is the bootstrap sample size used for estimating model selection probabilities; the sample size used for estimating candidate models is 8000. Second, performing model selection on TSS data observed at multiple time points (e.g. 0, 1.5, 5, . . .) is (in our opinion) beyond the scope of this paper (but probably well within the scope of a follow-up paper). Therefore, to address the sample size comment, we added results to Supplemental Information to show what happens for a smaller sample size (e.g. 4000). Also, as a reminder, Table 3 already showed copula results for smaller sample sizes of 3000 and 6000, respectively. To address the issue of another time point, we also show (in Supplemental Information) results from the analysis of TSS data observed at a later time (e.g. 4.5).

Comment 2 - Moreover, the manuscript should add a figure that shows how the data are fitted by different models.

Response to Comment 2 - This too is an excellent point. We have added Table 1 to Results showing the cost of each candidate model for the “ground truth” MEDIUM scenario.

Comment 3 - Also, there should be some brief discussions on model selection based on the other single-cell expression data (such as distribution data of gene products copy numbers: Mathematical Biosciences 345 (2022) 108780, PLoS Comput Biol 20(5): e1012118)

Response to Comment 3 - Both papers (which we have now cited in Discussion) deal with stochastic gene transcription/expression models with intrinsic stochastic noise, and such models cannot be described by ordinary differential equations. These papers address the issue of estimating parameters to fit the full probability distributions, but they do so for the steady state condition, in sharp contrast to the time-dependent evolution considered by us. The progress they make is possible because they have an exact solution to the telegraph model. Generalizing our approach to stochastic models poses formidable challenges on many levels. The experimental determination of the probability distribution is difficult; and the numerical simulation of models more complicated than the telegraph model using the Gillespie algorithm with sufficient statistical reliability to use methods to extract model parameters, the necessary first step before model selection, is itself extremely difficult. Al- though using approximations that truncate the moment expansion of the Master Equation to find low-order moments is the best that can be done [see Lu ̈ck, A. & Wolf, V. General- ized method of moments for estimating parameters of stochastic reaction networks. BMC Syst. Biol. 10, 98 (2016)], this approach is not perfect since the approximations introduce another potential layer of uncontrolled errors.

We have addressed this issue by stating clearly that our approach “ is not designed to handle intrinsic noise.”

Comment 4 - A typo: Line 66 “minus entropy). and since entropy depends only on f, it suffices (for the purpose of model”, period should be deleted.

Response to Comment 4 - The period in question has been replaced by a comma.

Reviewer 2 Report

Comments and Suggestions for Authors

See attached report.

Comments on the Quality of English Language

See attached report

Author Response

Comment 1 - Readability and Clarity: The manuscript’s readability is poor overall. The au- thors introduce numerous terminologies without sufficient explanation or accompanying mathematical expressions. For example, the term “the weighted sum of squared differ- ences between the first and second moments of y and xi ” is mentioned without elabo- ration. Additionally, in the “Data Description” section, the candidate models h1, h2 and h3 are referenced without any explicit mathematical expressions, leaving readers unclear about the models being analyzed.

Response to Comment 1 - We have taken this comment seriously, and as such to improve clarity and readability, we have re-written several parts of several sections (e.g. the first paragraph of Methods, most of Data Description, and subsections 5.1 and 5.2 in Discus- sion). Also, we have given more details about the weight matrix and provided additional references for interested readers. Finally, mathematical expressions for candidate models h1, h2, and h3 will typically not be available, and it is our ability to perform model selection in the absence of such information (i.e. in the absence of a likelihood function) that makes our approach novel.

Comment 2 - Reference to GMM-based Approach: In the abstract, the authors claim to have recently developed a GMM-based approach for parameter estimation using TSS data. However, this result is not substantiated in the references provided.

Response to Comment 2 - While the GMM based approach described in Wu et al. 2023 was primarily used to estimate parameters from signaling networks with deterministic ki- netics (ODEs) and extrinsic noise, the cited publication does illustrate several key points: (1) we can construct an optimal weight matrix from TSS data, (2) our particle swarm optimization routines are fast and efficient, and (3) we have the ability to estimate rate constants from the joint analysis of multiple snapshots observed over time. In addition, we have used our GMM based approach with remarkable success on coupled sets of linear ODE’s, systems with up to 10 measured proteins, and carefully constructed systems with both intrinsic and extrinsic noise. Furthermore, we have publicly available software for our GMM-based estimation procedure, and our software has been used by us and by others in the field. With all this said, we have re-organized the ideas in the Abstract to de-emphasize the ”development” of our GMM based approach, and instead we focus on the reality that we (and others) have been using our GMM based approach successfully for several years now.

Comment 3 - ...minimizing the above expression does not necessarily ensure that the KL divergence equals zero.

Response to Comment 3 - To be clear, if the “ground truth” is one of the candidate models under consideration, then that candidate model will minimize

CE(f(y) k hi(y | i)) Ef log hi(y | i) .

and the KL divergence between that candidate model and f will be zero. This is an important theoretical result that motivates much of our cross-entropy approach. However, since cross-entropy and KL divergence can not be computed, the best one can ever do (in practice) is to minimize an approximation. When the candidate models have likelihoods with mathematical expressions, the minimization over AICc (or AIC) often does the trick. By contrast, we developed ACE to handle situations where the likelihoods for candidate models do not have mathematical expressions.

Comment 4 - Mathematical Expressions for ODEs: In line 101 of page 3, the authors describe a set of coupled nonlinear ODEs governing the deterministic dynamics of single-cell protein abundances over time (di), where each di has reaction rates dependent on parameters i. However, the authors do not provide the explicit mathematical expressions for these ODEs or the parameters i.

Response to Comment 4 - Mathematical expressions for ODE’s were already given in Appendix A. Now we have added a forward reference to Appendix A in the text, and we have added the explicit values of all reaction rate constants in Appendix A as well.

Comment 5 - Assumptions about Variance-Covariance Matrix: In lines 150–155, the authors assume that the variance-covariance matrix of a multivariate normal random vector has off-diagonal elements equal to zero. Readers would benefit from an explanation of this assumption and its implications, as it might not always hold in practical scenarios.

Response to Comment 5 - This is an excellent question, and we re-wrote the last paragraph of Methods to address it (i.e. to describe the somewhat complicated AICc situation better). While we agree that the assumption of independence might not always hold in practice, we also believe that a thorough pursuit of the implications when this assumption is violated is beyond the scope of this paper, especially since our proposed approach (ACE) does not make any assumptions about normality or independence.

Comment 6 - Number of Candidate Models: In the “Methods” and “Data Description” sections, the authors restrict their analysis to three candidate models. It would be informative to discuss whether the proposed approach can handle more than three models.

Response to Comment 6 - In principle (subject of course to practical computational con- constraints), ACE can handle any number of candidate models, and we added a sentence to the last paragraph in Data Description that makes this point explicitly.

Comment 7 - Bootstrap Model Selection Probabilities: In Table 2, the authors present the probabilities of selecting candidate models based on 1,000 bootstrap resamples. Based on my understanding, these probabilities are calculated as the proportion of correctly selected models out of 1,000 samples. For instance, in the “ground truth” LARGE scenario, the ACE method selected the correct model 100% of the time. Thus, the values in columns 2–4 of Table 2 should sum to 1.

Response to Comment 7 - For each row of Table 2, the values should some to 1; and they do.

Reviewer 3 Report

Comments and Suggestions for Authors

The authors proposed an entropy based model selection method involving ordinary differential equations (ODEs) for single cell time-stamped (TSS) data, and assessed the performance of the model on a simulated dataset.

A major issue of the study is that there is no control methods used in the data analysis. The authors only applied their own method on simulated datasets in order to claim the advantage, which is not fully convincing since no benchmark methods are included for comparison. In addition, they only applied the entropy methods on simulated dataset. Of course, the model selection results appear perfect, as indicated in Table 2. If you conduct data analysis on real single-cell TSS data, will you still observe such a pattern since the “ground truth” is not necessarily known in advance in real data.

The entropy based model selection methods are not new. Please refer to the review paper in Wu et al and provide a more thorough discussion. In particular, the entropy based model selection methods reviewed in Wu et al. have all been applied and validated on real data analysis. I am wondering if the authors can demonstrate the advantage of their method on a case study?

Besides, details regarding the convergence of the entropy algorithm are lacking. For example, what is the stopping criterion adopted in the algorithm? Specifically, since the computation is based on approximation, monitoring the changes of the criterion value during optimization should be provided. Does the proposed algorithm lead to monotone changes (monotone increasing or decreasing) of the criterion value? Normally, even with ideal convergence performance on simulated data, it may not necessarily be carried over to real data analysis.  

References:

Wu, C., Li, S., & Cui, Y. (2012). Genetic association studies: an information content perspective. Current genomics13(7), 566-573.

Author Response

Comment 1 - A major issue of the study is that there is no control methods used in the
data analysis...

Response to Comment 1 - The Reviewer makes a good point; however, our paper is is
largely a “proof of principle paper” as opposed to a methods comparison paper. In fact,
we are suggesting quite strongly that AICc could complement ACE, so these two different
approaches may serve the research community better by working together, instead of com-
peting against one another. Moreover, the next logical step from our findings here is to
show that ACE can handle TSS data simulated with extrinsic and intrinsic noise. Once this
hurdle has been crossed, some sort of a methods comparison paper may be appropriate.

Comment 2 - The entropy based model selection methods are not new...

Response to Comment 2 - As per the Reviewer’s request, we have included a citation
to Wu et. al (2012) in Introduction highlighting the utility of mutual information in cer-
tain contexts. Furthermore, while we agree that entropy-based model selection methods
are not new, it is also true that there is essentially no statistically rigorous way to per-
form model selection without a mathematical expression for a likelihood. Nevertheless,
we have succeeded in doing just that! By removing the need for bias correction (i.e. using
a split-sample approach), and in the absence of a likelihood, by constructing asymptoti-
cally consistent estimators of (1) candidate model parameters (i.e. using the GMME in
place of the MLE), and (2) each candidate model likelihood (i.e. using a copula-based
estimate of each multivariate density), we have shown that model selection from TSS
data is possible. Specifically, time-stamped snapshots of a dynamic process can be used
to inform researchers about mechanistic models underlying the time-evolution of protein
abundances, in the absence of accurate mathematical expressions for the likelihoods of
each candidate model, and despite never once observing a single trajectory.

Comment 3 - Besides, details regarding the convergence of the entropy algorithm are lack-
ing. For example, what is the stopping criterion adopted in the algorithm?

Response to Comment 3 - We do not use an algorithm to compute ACE; hence, there is
no stopping criterion. We refer to ACE as an approximation because we are replacing the
MLE (which cannot be computed) with the GMME, and because we are replacing the un-
known likelihood with a copula based estimate. Conceptually, this is somewhat analogous
to using a limiting distribution to approximate a p-value, as opposed to computing an exact
p-value, which would require knowledge of the finite sampling distribution. Table 5 shows
that our copula-based log-likelihoods are within 1% of the true log-likelihoods for sample
sizes of 3000 and 6000. This suggests (for the simulated data considered here) that the
accuracy of ACE is at least comparable (if not better), since our ACE results are based on
a larger sample size.

Round 2

Reviewer 2 Report

Comments and Suggestions for Authors

No further comments

Reviewer 3 Report

Comments and Suggestions for Authors

Thank you. No further comments.